# Isoginkgetin—A Natural Compound to Control U87MG Glioblastoma Cell Growth and Migration Activating Apoptosis and Autophagy

**DOI:** 10.3390/molecules27238335

**Published:** 2022-11-29

**Authors:** Maria Antonietta Oliva, Sabrina Staffieri, Massimo Sanchez, Antonietta Arcella

**Affiliations:** 1IRCCS Istituto Neurologico Mediterraneo NEUROMED, Via Atinense 18, 86077 Pozzilli, Italy; 2Istituto Superiore di Sanita, 00161 Rome, Italy

**Keywords:** glioblastoma (GBM), isoginkgetin (Iso), temozolomide (TMZ), autophagy, apoptosis, PARP

## Abstract

Isoginkgetin (Iso) is a natural bioflavonoid isolated from the leaves of *Ginkgo biloba*, this natural substance exhibits many healing properties, among which the antitumor effect stands out. Here we tested the effect of Iso on the growth of U87MG glioblastoma cells. Growth curves and MTT toxicity assays showed time and dose-dependent growth inhibition of U87MG after treatment with Iso (15/25 µM) for 1, 2, and 3 days. The cell growth block of U87MG was further investigated with the colony formation test, which showed that iso treatment for 24 h reduced colony formation. The present study also aimed to evaluate the effect of Iso on U87MG glioblastoma cell migration. The FACS analysis, on the other hand, showed that treatment with Iso 15 µM determines a blockage of the cell cycle in the S1 phase. Further investigation shows that Iso treatment of U87MG altered the protein pathways of homeostasis including autophagy and apoptosis. The present study demonstrated, for the first time, that Iso could represent an excellent adjuvant drug for the treatment of glioblastoma by simultaneously activating multiple mechanisms that control the growth and migration of neoplastic cells.

## 1. Introduction

Isoginkgetin (Iso) is a natural bioflavonoid isolated from the leaves of *Ginkgo biloba* [1], which has manifested numerous healing properties, among which the anti-inflammatory, antioxidant, and anti-tumor properties certainly stand out. For thousands of years, *Ginkgo biloba* L., a famous Chinese herb, has been used to treat respiratory ailments such as asthma and bronchitis [2]; it was also famous for neuroprotective activity [3]. Furthermore, *Ginkgo biloba* L. determines an improvement of peripheral cardiovascular and vascular disorders, as well as influencing the platelet aggregation acting as a platelet antiplatelet [4]. A large number of documented studies have shown that G. biloba (EGB) leaf extracts show inhibitory effects on various types of cancer cells [5]. There is also experimental evidence that Iso, Gingko’s flavonoid extract, damages the extracellular matrix in cells by altering the production of metalloproteinase 9 (MMP-9), a protein known for modulating tumor metastasis and invasion. In fact, it is known that in human fibrosarcoma [6], the metalloproteases (MMP-9) is activated by the tissue metalloproteinase 1 (TIMP-1) and the inhibiting factor of the nuclear enhancer of the light chain kappa of activated B lymphocyte’s “NFκB” signaling [6,7].

Gingko biloba extract also has an inhibitory growth effect on MCF-7 and MDA-MB-231 human breast cancer [8], this property in particular motivated us to study its effect on glioblastoma cells. Glioblastoma multiforme (GBM) is the most aggressive and most frequent neoplasm originating from the cerebral nervous system. Its average incidence (i.e., the number of new cases per year) in Italy is eight cases per 100,000 inhabitants and represents 54% of all diagnosed gliomas with a median survival of about 18 months [9]. GBM is characterized by high neo-angiogenesis, pronounced mitotic activity, cellular heterogeneity, high rates of proliferation, and necrosis. The presence of cancer stem cells, capable of proliferating and generating glial neoplastic cells, contributes to the poor prognosis of patients with GBM [10], whose average survival is approximately 12 months from diagnosis. The therapeutic standard for all patients under 70 years of age and with good performance status after surgery or surgical biopsy is the combination of temozolomide 75 mg/m^2^/day for the duration of the entire radiotherapy (60 Gy/30 fractions) for a total of 7 weeks. This is followed by 6 cycles (which in some selected cases can be extended to 12 cycles) of adjuvant temozolomide at a dosage of 150–200 mg/m^2^/day for 5 days repeated every 28 days [11]. To date, clinical studies on the use of brain implant devices in the area of cancer, the so-called WAFERS of BCNU (carmustine) [12], an alkylating chemotherapeutic agent, should not be considered as a first-choice therapeutic option, given their little impact on survival.

The recurrence of GBM is unfortunately an almost inevitable event; therapeutic options range from the possible re-surgery to the resumption of treatment with temozolomide if there has been a sufficiently long interval of at least a few months or the use of chemotherapy regimens such as PCV (procarbazine, carmustine, and vincristine) or nitrosureas such as fotemustina. In conclusion, GBM patients have few therapeutic options, especially after failure of the initial standard treatment; however, there are many ongoing studies that are evaluating the use of new drugs. Furthermore, the intratumoral heterogeneity of GBM complicates the clinical outcome of the therapy [13]. The reconfiguration of the diagnostic criteria with the introduction of the WHO classification of CNS tumors in 2021 represented a first decisive change, which stratifies the tumors in molecularly different subsets towards a personalized therapy [14]. However, despite the efforts to classify GBM, patients currently have few therapeutic options, especially after the failure of the initial standard treatment. The ineffectiveness of chemotherapy agents can be attributed to the challenges of overcoming the blood–brain barrier. A growing body of evidence indicates that natural substances have shown high anticancer activity, which can be used successfully in the treatment of GBM as adjuvant therapy [15,16]. Relevant previous studies have shown valid inhibitory effects in vitro and in vivo of natural substances such as lactoferrin and aloe emodin [17,18]. The limit of these adjuvant substances is that after the scientific demonstration of the efficacy of the substance in vitro and in vivo, it is difficult to access to clinical trials that are fast, excessively complex for patient recruitment and too expensive. Here, we report the effect of isoginkgetin treatment on the continuous U87MG glioblastoma cell line. We found that Iso treatment at very low concentration (15 and 25 µM) determines an inhibition of growth and a reduction in migration in U87MG glioblastoma cells, two very interesting, combined effects.

## 2. Results

### 2.1. Cytotoxic Effect of Isoginkgetin on U87MG Glioblastoma Cells: Dose-Response and Time-Course

The cytotoxicity of Iso in glioblastoma was assessed using a continuous glioblastoma U87MG cell line and estimating the IC50 of Iso at 24, 48, and 72 h of treatment. As is evident in Figure 1B, the IC50 of Iso on U87MG cells decreased from 24.75 ± 13.7 µM (14.0 ± 0.5 µg/mL) at 24 h to 10.69 ± 3.63 µM (6.0 ± 0.5 µg/mL) at 72 h. Considering the strong cytotoxicity of this molecule, U87MG cells were treated daily with different concentrations of Iso (10, 15, and 25 µM) at different times (24, 48, and 72 h).

We observed that Iso treatment resulted in statistically significant time and dose growth inhibition of U87MG, as assessed by cell count (Figure 1C). According to the trend of the growth rate inhibition, the Iso reduced the viability of U87MG in a time and dose-dependent mode (Figure 1D), determining the maximum inhibitory effect at 72 h of treatment with 25 µM. The effect of Iso on U87MG cell proliferation was further investigated by colony formation assay. Figure 1E shows that the number of colonies decreased by 38% and 57%, respectively, in U87MG treated with Iso 15 and 25 µM compared to the control cells.

Iso determined a gradual morphological change in U87MG glioblastoma cells; the cells began to lose their characteristic tapered lance shape after 48 h of treatment to reach a rounded morphology. To further highlight these morphological changes, we performed immunohistochemical staining with an anti-vimentin antibody. As shown in Figure 1F, treatment with ISO 15 and 25 mM determines a redistribution of vimentin, which no longer appears distributed in the extensions but are thickened in a roundish shape (Figure 1F).

### 2.2. FACS Analysis

To investigate the effects of Iso on the cell cycle, U87MG cells were treated for 24, 48, and 72 h with the lower dose of Iso (15 μM). FACS analysis of U87MG glioblastoma cell control samples and treated with Iso 15 μM at different times (24, 48, and 72 h) shows a significant decrease in the S-phase percentage at 48 and 72 h (Figure 2). We observe an inhibition of the cell cycle; in particular, a decrement of S phase dose and time dependency is evident. The figure shows a trend of how the cell cycle is modulated: this trend is not yet significant, probably due to the short duration of the treatment. It is possible to hypothesize that damage to mitosis can occur from a longer ISO treatment, followed by a G2-phase enrichment of cells.

### 2.3. Iso Treatment Affects Migration of GBM Cells

The results of the scratch test show that Iso-treated cells repaired the scratch slower than untreated cells, on average percentage of closure of the scratch area of about 25% compared to 90% of the control cells (Figure 3A).

The migration assay in chamber slides also showed and confirmed, after 24 h of treatment of U87MG cells with Iso 15 and 25 µM, a decreased cell migration by 30% and 45%, respectively. We verified the phosphorylation state of the AKAP and ERK1/2 protein by Western blot after treatment with iso at different times. Figure 3D,E shows a decrease in phosphorylation of both AKT and ERK1/2 proteins after 24 h of treatment. We studied AKT and ERK phosphorylation because phosphorylation and de-phosphorylation have the ability to activate an extracellular signal that also regulates dynamic cell migration.

### 2.4. Proteolytic Activity of Iso on Metalloprotease in U87MG Glioblastoma Cells

To evaluate the proteolytic activity of MMPs in U87MG cells, under treatment with Iso, a zymography assay was performed using polyacrylamide gel (10%) containing 0.1% bovine (A) or porcine (B) gelatin. Molecular weights are shown on the left. The amount of secreted protein used in the gel, for all samples treated for 24 h with Iso 15 and 25 μM, was 150, 300, and 600 μg of proteins, respectively. As shown in Figure 3C, treatment with Iso at both concentrations results in a decrease in proteolytic activity, demonstrated by accumulation of MMP9 proteases in cancer cells, and also by decreased proteolytic fractions of 72 kDa (proMMP2) and active MMP2 (66 kDa).

### 2.5. Human Apoptosis Signaling Pathway Array

To investigate the mechanisms involved in ISG-induced blockade of cell proliferation, we performed an antibody array with two human apoptosis signaling pathway arrays (Figure 4). This assay, with immobilized antibodies, is used for protein examination in advantage that numerous proteins can be monitored in parallel. Therefore, we used a microarray with 19 proteins involved in the apoptotic pathway; two mirror membranes were used. On one membrane we hybridized protein lysates from control U87MG cells, on the other the protein lysates of cells were treated with Iso 25 µM for 48 h. The results are shown in Figure 4A, the proteins in question are BAD and caspase 3, cells with Iso for 48 h that determined an increase in the NFKBP65, which cleaved PARP, P38, P53. The expression values of each single protein were calculated by comparing the total expression with the expectation values of a housekeeping gene. The results of the array suggested the treatment of U87MG glioblastoma levels of some core proteins in the regulation of the apoptosis process (Figure 4A). To substantiate these results, we carried out Western blots with anti-caspase 3 antibodies anti-caspase 8 and anti PARP on lysates from U87MG cells treated with Iso 25 µM after 24, 48, and 72h. The results show that PARP is already activated after 24 h to reach maximum activation levels after 48 and 72 h as demonstrated by the increase in the cleaved band (89 kDa) (Figure 4C). A strong increase was also detected for the caspase 3-cleaved bands 45 kDa (Figure 4D).

### 2.6. Isoginkgetin and Autophagy

Another mechanism that cooperates in inhibiting the growth of U87MG cells after Iso exposition is the autophagy. The treatment of U87MG cells with Iso 15 and 25 µM determines an alteration of the autophagy protein. It is known that during autophagy the cytosol form of LC3BI is modified in the active form LCB3-II [19], which can interact and conjugates with phosphatidylethanolamine assembling in LC3–phosphatidylethanolamine conjugate (LC3-II), which is recruited to autophagosomal membranes [17]. This also occurs during the treatment of U87MG with Iso. The Western blot (Figure 4E), in the lysates from cells treated with ISO 15 and 25 µM after 48 h and 72 h, shows a decrease in the band relative to LC3BI (16 kDa) in favor of an increase in the LCB3-II band. Moreover, the level of P62 was modified to undergo Iso treatment and showed a time- and dose-dependent inhibition at both 24 and 48 h. Figure 4F shows that the band of 62 kDa decreased after treatment at all times (24, 48, 72 h) and at all concentrations (15 and 25 µM).

## 3. Discussion

Although progress has been made in the classification of brain tumors and new molecular targets have been identified, glioblastoma remains a disease with a poor prognosis and the therapeutic resources remain limited to the Stupp protocol: temozolomide and radiotherapy. This pushes more and more to study natural substances that can act in conjunction with chemotherapy drugs, trying to act on growth of neoplastic cells in a cooperative way, in order to stop their proliferation through multiple mechanisms; this will be able to overcome the variegated response of glioblastoma to canonical treatments. In this study, we wanted to evaluate the effect of the Iso on U87MG glioblastoma cells. As described above, isoginkgetin is a natural substance from the leaves of *Ginkgo biloba*. Numerous properties are attributed to the leaves’ extract, among which, certainly, the anti-inflammatory and antioxidant ones stand out. More specifically, the anti-inflammatory action attributed to this plant is traceable to the ginkgolides contained in it. In particular, some studies have shown that ginkgolide B is able to inhibit the activity of PAF (platelet-activating factor) through the antagonization of its receptor. The platelet activation factor, in fact, plays an important role in inflammatory processes and variations in vascular permeability.

Among these stands a relatively recent study (2001), which highlighted how ginkgo extract is able to increase the efficacy and tolerability of 5-fluorouracil (5-FU) in patients with colon cancer straight from refractory to treatment with only 5-FU [20]. This clinical study drove us to investigate the effect of *Ginkgo biloba* isoginkentine leaf derivatives on U87MG glioblastoma cells.

In the first experiments, the effect of Iso on cell growth was analyzed; in particular, U87MG growth curves and MTT toxicity assays after treatments with Iso cells showed a time (24, 48, 72 h)- and dose (15, 25 µM)-dependent inhibition of growth. Iso treatment modified glioblastoma morphology, the cell widespread with extensions become rounded and devoid of extensions after 48 h of treatment with 15 or 25 µM Iso. This morphological change is further highlighted by immunofluorescence staining with anti-vimentin antibody, the treatment with Iso 15 and 25 µM determines a redistribution of the vimentin, which no longer appears distributed in the extensions but thickened in a roundish cell similar to a rounded apoptotic-like cell. These results indicate that the link between the vimentin protein and DNA may be functional to the protein itself to grow and move cancer cells [21]. Growth inhibition of U87MG glioblastoma cells was further investigated by colony formation assay after 24 h of treatment with Iso 15 µM. The number of colonies was reduced by approximately 35% after 15 days in GBM cells treated with 15 µM Iso, and about 60% with 25 µM Iso compared to control cells not subjected to treatments (Figure 1E). FACS analysis of cell samples treated with Iso 15 μM, at different times (24, 48, and 72 h), shows a decrease in the percentage of cells in S-phase as early as 24 h after treatment, becoming significant at 48 and 72 h of treatment. The number of cells in S phase decreases with time exposition.

The migration of glioma cells within the brain parenchymal represents one of the mechanisms that makes glioblastoma one of the most aggressive and lethal tumors. In this paper we reported a very interesting effect of the treatment of glioblastoma cells with Iso represented by a reduction in cell migration as evidenced by a scratch test. Iso-treated U87MG glioblastoma cells repaired the scratch slower than untreated cells, on average percentage of closure of the scratch area of about 25% compared to 90% of the control cells. The migration assay in-chamber slides also strengthened and confirmed that after 24 h of treatment with Iso 15 and 25 µM, U87MG cell migration decreased by 30% and 45%, respectively. The proteolytic activity of metalloproteinase (MMPs) influenced tumor progression: primary tumor growth, angiogenesis, extravasation and invasion of neoplastic cells, migration, and invasion of metastatic cells in the secondary organ. In particular, it has been previously demonstrated in breast cancer that MMPs can also suppress tumor progression through the decreasing of proteolytic activity [22,23]. The U87MG zymography assay showed that treatment with 25 µM Iso for 24 h determined a decrease in the proteolytic activity of MMP2. Moreover, in U87MG glioblastoma cells, treatment with Iso determined an intensification of the proteolytic activity of the metalloproteases after 24 h of treatment; the levels of MMP-2 were reduced with an increase of MMP-9 form. This proteolytic activity induced by Iso-reduced cell migration could make Iso an excellent remedy for blocking the dissemination of neoplastic glial cells in the brain as a control strategy in the development of relapse, which complicates the patient’s prognosis, leading to a quick death. Finally, we investigated the mechanisms involved in the growth block by studying the main phenomena that control cell growth homeostasis: apoptosis and autophagy. Autophagy and apoptosis represent distinct physiological processes within the cell. Apoptosis is the best-known form of programmed cell death (Type I cell death). This occurs through the activation of catabolic enzymes by signal cascades, which leads to the rapid breakdown of cells structures and organelles [24]. In some circumstances, therefore, apoptosis and autophagy can exert synergistic effects, while in other situations autophagy can be induced only when apoptosis is suppressed [25]. In general terms, it seems that similar stimuli can induce both autophagy and apoptosis, so it is possible to find a mixed phenotype. In fact, in some settings, autophagy and apoptosis seem to be interconnected and the idea that “molecular switches” exist between the two processes has been hypothesized [7,26]. Similarly, in the present study, Western blot results pointed out an increase in cleaved band 89KD9 PARP (PARP activated) and a strong increase in caspase 3-cleaved bands after Iso treatment. The activation of apoptosis was also endorsed by DNA fragmentation evidenced in the U87MG Iso-treated by tunnel assay. The contemporary role of the presence of the autophagy process in Iso-treated U87MG glioblastoma cells was manifested by an increased in active band LCB3-II (16 kDa) and altered the levels of p62. This confirms that Iso may also activate a protagonist of the autophagy process. The adapter protein p62, also called sequestosome 1 (SQSTM1), is a ubiquitin-binding scaffold protein that colocalizes with ubiquitinate; generally, p62 accumulates when autophagy is inhibited, and decreases in level when autophagy is induced. Monitoring of the autophagic degradation of p62/SQSTM1 represents the indicator of the autophagic process. In our contest the decrease in p62 levels, after Iso 15 and 25 μM treatment, which contrasts with the intensification of the autophagic process, can be explained as the impairment of cell protein degradation machinery of cancer cells [27]. Iso could represent an excellent adjuvant for glioblastoma care, together with traditional temozolomide treatments and radiotherapy.

## 4. Conclusions

Considering the extreme intra- and intertumoral heterogeneity of glioblastoma and the innate or acquired resistance of high-grade brain tumors to conventional therapy, the present study aimed to evaluate the effect of Iso, a natural substance, as a single inhibitor agent on the growth, clonogenic potential, and migration of U87MG. Furthermore, an attempt was made to identify the main mechanisms underlying the inhibition of growth induced by Iso, highlighting that Iso has effects on the blocking of cell growth through a reactivation of apoptosis and autophagy. The most interesting future research, certainly, could be investigating the ability of Iso to act in synergy with temozolomide-bypassing drug resistance mechanisms, preventing therapy failure and reactivating the homeostatic mechanism in cancer cells.

## 5. Materials and Methods

### 5.1. Cell Culture

The continuous human glioblastoma cell line U87MG was purchased from Sigma Aldrich Collection (LGCPromochem, Teddington, UK); cells were growth in Dulbecco’s Modified Eagle’s Medium supplemented with 10% fetal bovine serum, 2 mmol/L-glutamine, 100 IU/mL penicillin, 100 µg streptomycin, at 37 °C, 5% CO_2_ and 95% of humidity. In vitro treatments were performed with pure molecule Isoginkgetin (Iso) from MedChemExpress, South Brunswick, NJ, USA, and Temozolomide (TMZ) from Sigma Aldrich (St. Louis, MI, USA).

### 5.2. Determination of Half-Maximal Inhibitory Concentration (IC50) of Isoginkgetin in U87MG Cells

To estimate the IC50-values of Isoginkgetin in U87MG at 24, 48 and 72 h, cells were plated in 96-well plates (5 × 10^3^ cells/well) and treated with Iso at different concentration (0.001, 0.01, 0.1, 1, 10, 25 and 50 µM), by only one induction. The IC50-values were calculated using GraphPad Prism (GraphPad Software Inc., San Diego, CA, USA).

### 5.3. Proliferation Assay

To evaluate the effect of isoginkgetin on U87MG proliferation, cells were plated in 48-well plates (1 × 10^4^ cells/well) in DMEM 10% FBS and incubated at 37 °C, 5% CO_2_. The day after, cells were treated with Iso at three different concentrations (10, 15, and 25 µM) by only one induction; DMSO 0.1% was used as vehicle control. Cell count was performed at 24, 48, and 72 h of treatment using a Burker chamber.

### 5.4. Cell Viability Assay

U87MG cells were seeded in 96-well plates (5 × 10^3^ cells/well) and treated with Iso at three different concentrations (10, 15 and 25 µM) by only one induction; DMSO 0.1% was added in the control. At 24, 48, and 72 h of treatment, cell viability was evaluated by MTT assay. Briefly, 10 µL of 5 mg/mL MTT (3-(4,5-dimethylthiazol-2-yl)-2,5-diphenyltetrazolium bromide; Sigma-Aldrich) was added in each well with 100 µL of medium. After 1 h of incubation at 37 °C, formazan crystals were dissolved with 100 µL isopropanol/HCl 0.4% and the absorbance was measured at 570 nm with a plate reading spectrophotometer.

### 5.5. Immunofluorescence

To analyze the morphological change of U87MG induced by isoginkgetin, cells were plated in 8-well chamber slides (10 × 10^4^ cells/well) in DMEM 0.5% FBS. After 48 h the medium was replaced with DMEM 10% FBS and cells were treated with Iso 15 and 25 µM for 48 h. At the end of treatment, cells were fixed in 4% formalin pH 7.4 (Diapath, Martinengo, Italy) for 20 min and permeabilized with 0.1% Triton (Invitrogen, Carlsbad, CA, USA) for 30 min. After blocking with 10% horse serum (Vector) for 1 h, cells were incubated with a mouse monoclonal antibody anti-vimentin (prediluted; Roche Diagnostic, Mannheim, Germany) overnight at 4 °C. After washing with 0.025% PBS-Tween-20 (Sigma-Aldrich, St. Louis, Missouri, USA) cells were incubated with secondary antibody anti-mouse fluorescein (1:100; Vector) in 2% serum for 1 h at room temperature. The slide was counterstained with DAPI mounting medium (Vectashield) and analyzed with a fluorescence microscope (Axiophot 2 Zeiss, Gottigen, Germany) at 20× magnification.

### 5.6. Wound Healing Assay

To evaluate the effects of Iso on U87MG motility, cells were seeded into 6-well culture plates (5 × 10^3^ cells/well); when the cells reached confluence, a scratch was made by sterile 100 µL pipette tips and detached cells were washed with PBS 1×. After adding of DMEM 10% FBS, an induction with Iso 15 and 25 µM was performed. Cell migration was followed by observation at a phase contrast microscope (Evos, Life Technologies, Carlsbad, CA, USA) at 4× magnification. The scratch area was quantized by image analysis with ImageJ software.

### 5.7. Migration Assay

U87MG cells were seeded (15 × 10^4^ cells/well) in a serum-free medium in Boyden chambers (BD Falcon, Schaffhausen, Switzerland) immersed in 2 mL of FBS. After adhesion to the bottom of the chambers, cells were treated with Iso 15 and 25 µM and incubated at 37 °C for 48 h. The cells migrated through the filter were fixed with methanol for 5 min and stained with hematoxylin and eosin. The count of migrated cells was carried out by an optical microscope at 40× magnification in 5 different fields.

### 5.8. Clonogenic Assay

Colony formation assay was performed seeding U87MG cells in 6-well plates (10^3^ cells/well) in DMEM 10% FBS for 48 h. Cells were treated with Iso 15 and 25 µM for 24 h and medium was replaced every 3 days for 14 days. Cells were fixed with 4% paraformaldehyde solution for 5 min, washed with PBS 1× and stained with crystal violet 0.05% for 30 min. The colony count was carried out by an optical microscope at 40× magnification.

### 5.9. Zymography Assay

The effect of Iso on metalloprotease enzymatic activity was investigated with a Zymography assay. U87MG cells were plated in DMEM 0.5% FBS (4 × 10^4^ cells/well) for 48 h; at 70–80% confluency, FBS medium was replaced with FBS-free medium and the cells were induced with Iso15 and 25 µM for 24 h. After treatment, culture medium was collected and concentrated in a SpeedVac centrifuge (Thermo Scientific, Waltham, Massachusetts). Different concentrations of protein from culture medium (150 ηg, 300 ηg, and 600 ηg), in 10 µL of non-reducing sample buffer (4% SDS, 20% glycerol, 0.01% bromophenol blue, 125 mM Tris-HCl pH 6.8), were separated on a 10% polyacrylamide gel containing porcine gelatin (4 mg/mL; Sigma-Aldrich, Missouri, St. Louis, MO, USA). The gel was immersed for 30 min in washing buffer (2.5% Triton X-100, 50 mM Tris-HCl pH 7.5, 5 mM CaCl_2_, 1 µM ZnCl_2_) and stirred in incubation buffer (1% Triton X-100, 50 mM Tris-HCl pH 7.5, 5 mM CaCl_2_, 1 µM ZnCl_2_) for 24 h at 37 °C. Finally, the gel was stained with staining solution (40 mL methanol, 10 mL acetic acid, 50 mL H_2_O, 0.5 g Coomassie blue) for 30 min, rinsed with H_2_O and incubated in destaining solution (40 mL methanol, 10 mL acetic acid, 50 mL H_2_O) until bands were clearly seen. Areas of enzyme activity appeared as white bands against a dark blue background.

### 5.10. Cell Cycle Analysis by Flow Cytometry

U87MG cells were plated in DMEM 0.5% FBS (5 × 10^5^ cells/well) for 48h and treated with Iso 15 and 25 µM. At 24, 48, and 72 h of treatment, cells were washed in sample buffer (0.1% Glucose in HBSS without Ca^++^ and Mg^++^), resuspended in sample buffer and fixed in ice-cold 70% ethanol overnight at 4 °C. After centrifugation, cells were incubated in propidium iodide (PI) staining solution (50 µg/mL of PI) for 30 min at room temperature and analyzed within 24 h in a flow cytometer (Gallios Instrument, Beckman Coulter, Brea, CA, USA). Cell cycle analysis was performed by using Kaluza software for analysis v. 2.1 (Beckman Coulter).

### 5.11. DNA Fragmentation’s Detection by TUNEL Assay

U87MG cells were plated in 8-well chamber slides (10 × 10^4^ cells/well), in DMEM 0.5% FBS, for 48 h and treated with Iso 15 and 25 µM for 48 h. After treatment, cells were fixed in 4% methanol-free formaldehyde solution pH 7.4 for 25 min at 4 °C and washed twice with PBS 1×. DNA fragmentation was detected by TUNEL assay with commercial kit DeadEnd Fluorometric TUNEL System (Promega, Madison, WI, USA), following the manufacturer’s instructions. Cells were counterstained with DAPI mounting medium (Vectashield) and analyzed with a fluorescence microscope (Axiophot 2 Zeiss) at 40× magnification.

### 5.12. Human Apoptosis Signaling Pathway Array

The analysis of several proteins involved in the apoptosis pathway was carried out in U87MG control cells and treated with Iso 25 µM for 24 h using the human apoptosis signaling pathway array (RayBiotech, Norcross, GA, Peachtree Corners, GA, USA). Two antibody arrays were incubated in blocking buffer for 30 min at room temperature and then with protein samples (200 µg proteins in 1ml of blocking buffer) overnight at 4 °C. After washing in a specific buffer, each membrane was incubated with a detection antibody cocktail and then with HRP-anti-rabbit IgG for 2 h at room temperature, respectively. Two membranes were washed twice with specific buffer, incubated with chemiluminescence detection buffer for 2 min and analyzed with a ChemiDoc XRS imaging system (Bio-Rad Laboratories, CA, USA). Each membrane had a positive control to normalize the signal of all antibodies. The membrane with proteins from U87MG treated was considered as the “Reference Array” to which the other membranes were normalized too.

### 5.13. Western Blot Analysis of Proteins Involved in Apoptosis and Autophagy

U87MG cells were seeded in DMEM 0.5% FBS (5 × 10^5^ cells/well) for 48 h and treated with Iso 15 and 25 µM. A short-term treatment (15 min, 30 min, 1 h, 2 h, 4 h, 6 h, and 12 h) was performed for analysis of phosphorylated proteins (pAKT and pERK1/2) and a long-term treatment (24, 48, 72 h) for analysis of all other proteins (PARP, caspase 3, LC3 and P62). Proteins were extracted from U87MG cells with Triton X-100 lysis buffer (10 mM Tris-HCl, 1 mM EDTA, 150 mM NaCl, 1% Triton X-100, 1 mM NaF, 1 mM Na_4_P_2_O_7_, 1 mM Na_3_VO_4_, and 1x protease inhibitors). Protein lysates were resolved on 10–12% SDS-PAGE and transferred to PVDF membranes (Bio-Rad, California, USA) by electroblotting. Each blot was incubated for 1h at room temperature in 5% non-fat dry milk or bovine serum albumin (BSA) diluted in 1× Tris-buffered saline–0.1% Tween-20 and then was incubated overnight at 4 °C with primary antibody, diluted in 2% milk or BSA. After washing in 1 × TBS-0.1% Tween the membrane was incubated for 1 h at room temperature with a specific HRP-conjugated secondary antibody (anti-mouse or anti-rabbit 1:7000; Calbiochem, St. Louis, MO, USA) and then for 1 min with ECL (Amersham Pharmacia Biotech Italia Spa, Cologno Monzese, Italy). Blot was analyzed with a ChemiDoc XRS Imaging System (Bio-Rad) and the digital signals were quantified by Image Lab Software (Bio-Rad Laboratories). For analysis of phosphorylated proteins, we used two primary antibodies, anti-pERK1/2, and anti-pAKT, (Cell Signaling; 1:1000, Danvers, MA, USA) in 2.5% BSA, while for the proteins involved in apoptosis and autophagy, we used different primary antibodies (anti-PARP, anti-caspase 3, anti-LC3, and anti-P62), all from Cell Signaling, 1:1000 in 2.5% milk. For protein normalization, each membrane was incubated with a mouse monoclonal antibody, anti-beta-actin (Cell Signaling;1:1000) or a mouse monoclonal antibody anti-GAPDH (Cell Signaling; 1:1000).

### 5.14. Statistical Analysis

Data were expressed as means ± SEM of three individual experiments; statistical significance was determined by one-way ANOVA test and Dunnett’s multiple comparison test, considering *p*-values < 0.05 statistically significant, according to GraphPad Prism.

## Figures and Tables

**Figure 1 molecules-27-08335-f001:**
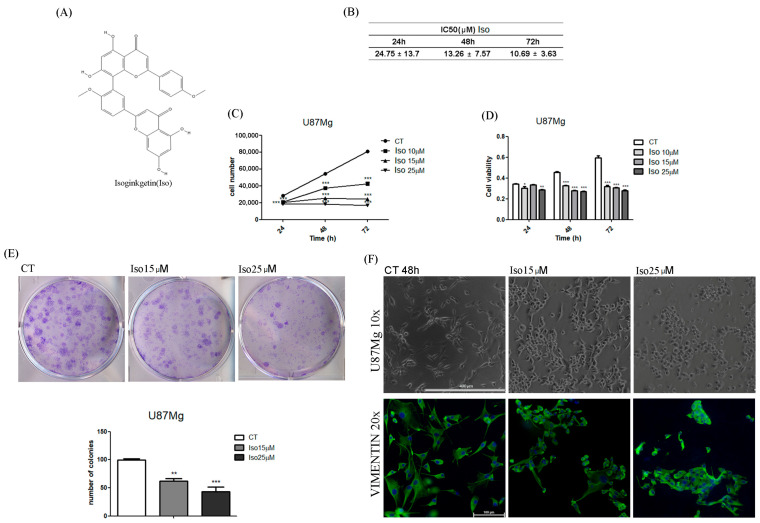
(**A**) Chemical structure of Isoginkgetin. (**B**) IC50-values of U87MG cells after 24, 48, and 72 h of treatment with Iso. (**C**) Time and dose-dependent growth inhibition of U87MG cells treated with Iso at different concentrations (10, 15, 25 µM). (**D**) Cell viability of U87MG cells after treatment with Iso (10, 15, 25 µM) for 24, 48, and 72 h. (**E**) Representative images and count of the colonies in U87MG at 12 days from treatment with Iso 15 and 25 µM. (**F**) Morphological changes of U87MG treated with Iso 15 and 25 µM for 48 h; (up) brightfield images acquired by a phase contrast microscope (Evos, Life technologies) at 10× magnification and (down) vimentin fluorescence images acquired by a fluorescence microscope (Axiophot 2 Zeiss) at 20× magnification. Values are the means ± SEM of three individual experiments: one-way ANOVA test, Dunnett’s multiple comparison test, * *p* values < 0.05. According to GraphPad Prism ** *p* value 0.001 to 0.01 (very significant), *** *p* value 0.0001 to 0.001 (extremely significant).

**Figure 2 molecules-27-08335-f002:**
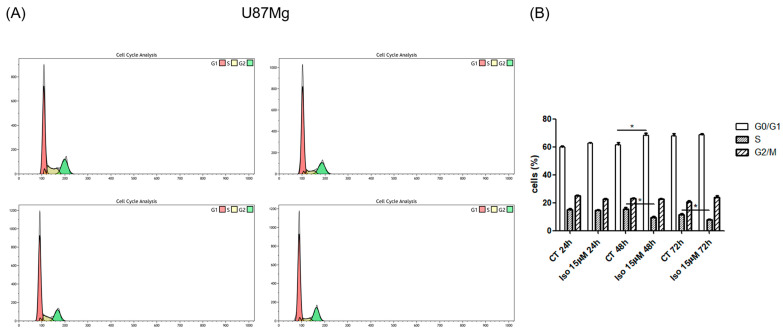
(**A**) Representative images of FACS analysis in U87MG cells at 48 and 72 h of treatment with Iso 15 µM. (**B**) Percentages of U87MG cells in the different phases of the cell cycle after treatment with Iso 15 µM: isoginkgetin causes a decrease of the cells in S phase and an increase in G2 phase at 48 and 72 h of treatment. Values are the means ± SEM of three individual measurements: unpaired *t*-test, one-way ANOVA test, Bonferroni’s multiple comparison test, *p*-values < 0.05. According to GraphPad Prism, * *p* value 0.01 to 0.05 (significant).

**Figure 3 molecules-27-08335-f003:**
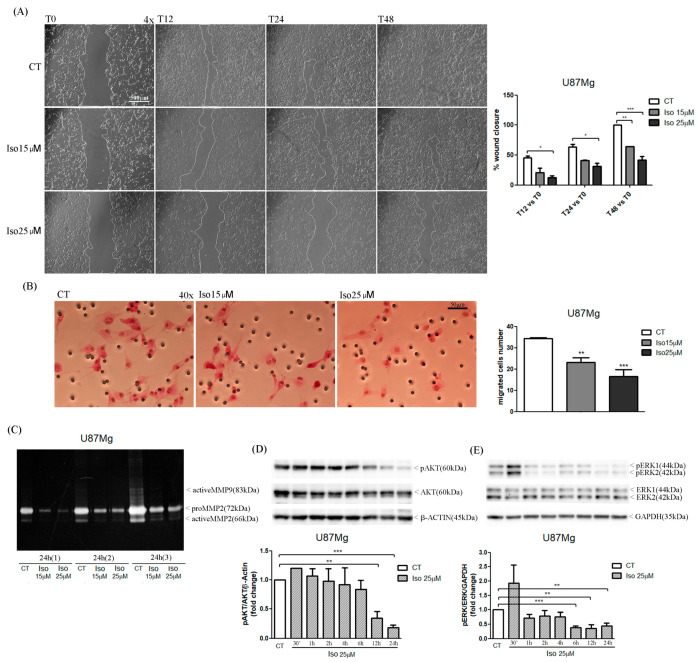
(**A**) Wound healing assay in U87MG treated with Iso 15 and 25 µM: the closure of the scratch area in treated cells is less than in control cells, at all observed times (scale bar 500 µm). (**B**) Migration assay in U87MG treated with Iso 15 and 25 µM for 48 h: the number of migrated cells in the treated U87MG is lower than in the controls (scale bar 50 µm). (**C**) Zymography assay in culture medium collected from U87MG after 24 h of induction with Iso 15 and 25 µM: in treated cells isoginkgetin reduces the enzymatic activity of MMP9 (83 kDa), proMMP2 (72 kDa), and activeMMP2 (66 kDa) compared to control cells. (**D**,**E**) Western blot analysis of pAKT and pERK1/2 in U87MG after a short-term treatment with Iso 25 µM: Isoginkgetin significantly reduces the levels of phosphorylated AKT only at 12 and 24 h oh treatment (**D**) and reduces the levels of phosphorylated ERK1/2 already at 6 h of treatment (**E**). Values are the means ± SEM of three individual experiments. Unpaired t-test, one-way ANOVA test, Bonferroni’s multiple comparison test, * *p* values < 0.05. According to GraphPad Prism, ** *p* value 0.001 to 0.01 (very significant), *** *p* value 0.0001 to 0.001 (extremely significant).

**Figure 4 molecules-27-08335-f004:**
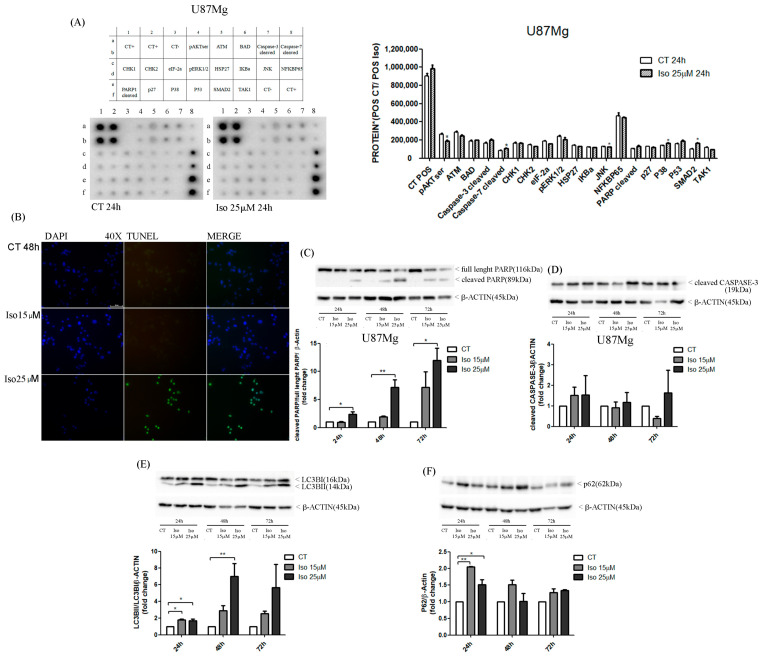
(**A**) Human apoptosis signaling pathway array in U87MG cells control and treated. (**B**) TUNEL assay in U87MG treated with Iso 15 and 25 µM for 48 h: isoginkgetin 25 µM causes a DNA fragmentation after 48h of treatment. (**C**,**D**) Western blot analysis of PARP and Caspase-3 in U87MG cells treated with Iso 15 and 25 µM for 24, 48 and 72 h: in the treated cells compared to the control there is an increase of cleaved PARP (**C**) and cleaved caspase-3 (**D**) at both concentrations of Iso and at all times. (**E**,**F**) Values are the means ± SEM of three individual experiments. One-way ANOVA test, Bonferroni’s multiple comparison test, *p* values < 0.05. According to GraphPad Prism, * *p* value 0.01 to 0.05 (significant), ** *p* value 0.001 to 0.01 (very significant).

## Data Availability

The raw data supporting the conclusions of this article will be made available by the authors upon request, without undue reservation.

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
