# Peer review of "Isoginkgetin—A Natural Compound to Control U87MG Glioblastoma Cell Growth and Migration Activating Apoptosis and Autophagy"

_molecules, 2022, doi:10.3390/molecules27238335_

Round 1
Reviewer 1 Report
This is promising work that has examined the anti-glioblastoma effects of Isoginkgetin and has yielded positive results. However, there are still some issues that need to be addressed.
1. I would suggest adding sufficient background to the research in the introduction, for example by mentioning some of the important advances currently made in the field of natural products against glioblastoma, what problems remain, and how the results of the authors' work will differ or advance from those of their predecessors.
2. The resolution of Figure 1 is too low for the reader to see the results of the experiment clearly. The scale in Fig. 1F is also not visible.
3. The resolution of Figure 2A is too low. Figure 2B shows that the proportion of cells in G1 phase increased and the S phase decreased after 48 h of compound treatment. However, cells showed a decrease in the S phase ratio after 72 h of compound treatment, but no significant change in either G1 or G2/M. The authors should mention this result and explain it in the description of the results.
4. The scale should be added to Fig. 3A, and the scale in Fig. 3B is not clear enough.
5. Line 117-119 “The results of the scratch test show that Iso treated cells repaired the scratch slower than untreated cells, on average percentage of closure of the scratch area of about 25% compared to 90% of the control cells (Figure 3 B, C).” Please check that the illustration cited here is correct.
6. Please check the spelling of the sentences in lines 122-123.
7. There are many proteins related to cell migration, why did the authors choose to detect AKT and ERK, I would like to suggest the authors add something to support this content.
8. Please make sure that the correct image is used for Figure 4A, as visually there is no significant difference in protein expression between the two protein microarrays.
9. The author does not seem to have mentioned Fig. 4B in the text, please confirm it. The magnification and resolution of this image is not satisfactory, and the scale is missing.
10. I would like to suggest that the authors add a flow cytometry assay to check if isoginkgetin induces apoptosis in U87MG glioblastoma cells.
11. I would like to suggest that the authors add a transmission electron microscope photograph as direct evidence that the compound induces autophagy.
12. In this study, the authors concluded that the tested compounds acted by inducing apoptosis and autophagy, so there should be direct evidence that the compounds induced apoptosis and autophagy. The title of the manuscript deserves further deliberation. Also, the spelling of the manuscript needs to be checked and revised.
Author Response
I and the other authors of the paper thank you for appreciating our study and for giving us valuable suggestions to further improve the presentation of our data:
This is promising work that has examined the anti-glioblastoma effects of Isoginkgetin and has yielded positive results. However, there are still some issues that need to be addressed.
- I would suggest adding a sufficient background to the research in the introduction, for example by mentioning some of the important advances currently made in the field of natural products against glioblastoma, what problems remain, and how the results of the authors' work will differ or advance from those of their predecessors.
We add in the introduction a background about the use of natural substances for glioblastoma therapy, thanks to your suggestion to meliorate this aspect.
- The resolution of Figure 1 is too low for the reader to see the experiment's results clearly. The scale in Fig. 1F is also not visible.
We meliorate the resolution and add a scale bar, in FIG. 1 We send also the original picture separately.
- The resolution of Figure 2A is too low. Figure 2B shows that the proportion of cells in the G1 phase increased and the S phase decreased after 48 h of compound treatment. However, cells showed a decrease in the S phase ratio after 72 h of compound treatment, but no significant change in either G1 or G2/M. The authors should mention this result and explain it in the description of the results.
As shown in Figure 2B the number of cells in the S phase significantly decreases both 48 and 72 h of Isoginkgetin treatment. The reduction of cells in the S phase is associated with a significant increase in the number of cells in G1 at 48 h of treatment but not at 72 h. This discrepancy could depend on a reduced cell growth shown by controls after 72 h in culture, due to a decrease in nutrients and/or an increase in cell concentration with consequent growth inhibition by cell contact. This slowdown in cell growth is suggested by the observation that at 72h the percentage of control and treated cells in G1 was comparable, reaching values ​​similar to those observed at 48 h in isoginkgetin-treated culture. The hypothesis of a growth slowdown is also suggested by a decrease in control cells in the S phase at 72 h compared to 48 h, and by an increase in the G2 cell fraction in treated cultures, although in both cases not significant. In any case, the time course experiment at 24, 48, and 72 h clearly demonstrate that isoginkgetin treatment for 48h induces a slowed cell cycle with a significant decrease of cell percentage in the S-phase and accumulation in the G1-phase.
- The scale should be added to Fig. 3A, and the scale in Fig. 3B is not clear enough.
We correct the scale in Fig 2A and 2 B..
- Line 117-119 “The results of the scratch test show that Iso-treated cells repaired the scratch slower than untreated cells, on the average percentage of closure of the scratch area of about 25% compared to 90% of the control cells (Figure 3 B, C).” Please check that the illustration cited here is correct.
The correct figure is 3A, thanks thank you for pointing out this inconsistency.
- Please check the spelling of the sentences in lines 122-123.
I correct in the test this point.
- There are many proteins related to cell migration, why did the authors choose to detect AKT and ERK, I would like to suggest the authors add something to support this content.
We study AKT and ERK phosphorylation because phosphorylation and de-phosphorylation have the ability to activate extracellular signals- that regulating also dynamic cell migration.
Mei-Ying Han, Hidetaka Kosako, Toshiki Watanabe, and Seisuke Hattori. Mol Cell Biol. 2007 Dec; 27(23): 8190–8204.doi: 10.1128/MCB.00661-07. Extracellular Signal-Regulated Kinase/Mitogen-Activated Protein Kinase Regulates Actin Organization and Cell Motility by Phosphorylating the Actin Cross-Linking Protein EPLIN
- Please make sure that the correct image is used for Figure 4A, as visually there is no significant difference in protein expression between the two protein microarrays.,
Figure 4A is correct, we used initially the protein array only to suggest how proteins changed after Iso stimulation. To clarify the specific protein involved in Iso cell growth inhibition we make a specific western blot for apoptotic markers (PARP, caspase III) and autophagic markers as LCII.
- The author does not seem to have mentioned Fig. 4B in the text, please confirm it. The magnification and resolution of this image are not satisfactory, and the scale is missing.
We meliorate resolution and add in the test and discussion Fig 4B.
- I suggest that the authors add a flow cytometry assay to check if isoginkgetin induces apoptosis in U87MG glioblastoma cells.
I think that the presence of apoptosis was consistently demonstrated by western blot increase of PARP cleaved and caspase-3 activations. Furthermore, the Tunnel assay (Fig. 4 b) shows DNA fragmentation in U87MG Iso treated.
- I would like to suggest that the authors add a transmission electron microscope photograph as direct evidence that the compound induces authophagy.
we were unable to carry out this experiment because we do not have the equipment and the specific skills. Furthermore, our study aims to demonstrate the inhibitory effect of Iso on glioblastoma cells just to hypothesize some mechanisms of action.
- In this study, the authors concluded that the tested compounds acted by inducing apoptosis and autophagy, so there should be direct evidence that the compounds induced apoptosis and autophagy. The title of the manuscript deserves further deliberation. Also, the spelling of the manuscript needs to be checked and revised.
We changed the title and revised the spelling in all manuscripts.
Reviewer 2 Report
The manuscript "Isoginkgetin a Natural Compound to Control U87MG Glioblastoma Cell Growth and Migration" is interesting to read. The study will be more valuable to researchers working in the field; thus, I appreciate the authors effort in conducting it. However, the manuscript might be considered for publication in Molecules if the following modifications are made and included in it.
1. I advise authors to modify the title as "Isoginkgetin - A Natural Compound Inhibits U87MG Glioblastoma Cell Migration".
2. Avoid I/We/Our throughout the manuscript. Instead use “The present/current study”
3. Revise the objective in Line 15. “The present study aimed to evaluate the effect of Isoginkgetin on U87MG glioblastoma cell migration”.
4. ISO or Iso, make it uniform in throughout the manuscript.
5. In the abstract, describe the conclusion in more detail and offer suggestions for the reader. so that readers will understand the overall purpose of this research after reading the abstract.
6. Only two of the introduction's references were recently published (10 and 11). Particularly when discussing "Glioblastoma," the introduction section needs to be updated with the most recent references that have been published in the last five years (12, 13 and 14).
7. Line 65-69, revise it. The readers will have trouble understanding the objective.
8. In figure 2, there was no ** p-value and *** p-value, please check the figure legends and remove it.
9. Similarly, in figure 4, there was no *** p-value, please check the figure legends and remove it.
10. The writing in the results section is excellent. Another issue I observed was the lack of explanation in the discussion. Some results were mentioned again in the discussion. Minimize it. I would advise the authors to refocus their discussion on each section rather than adding more background information about the literature so that it is clear how the research's findings fit into the overall framework of what is happening with glioblastoma at the moment. The methods section was well detailed. I appreciate it.
11. I advise authors to break up the conclusion section and provide a critical defence of their findings and observations. As a result, everyone will recognise the significance of this study. The conclusion must also address potential points of view. The author should stress the significance of this study piece.
Author Response
The manuscript "Isoginkgetin a Natural Compound to Control U87MG Glioblastoma Cell Growth and Migration" is interesting to read. The study will be more valuable to researchers working in the field; thus, I appreciate the authors effort in conducting it. However, the manuscript might be considered for publication in Molecules if the following modifications are made and included in it.
I and the other authors of the paper thank you for appreciating our study and for giving us valuable suggestions to further improve the presentation of our data:
- I advise authors to modify the title as "Isoginkgetin - A Natural Compound Inhibits U87MG Glioblastoma Cell Migration".
We modify the title in Isoginkgetin - A Natural Compound Inhibits U87MG Glioblastoma Cell Migration activating apoptosis and autophagy”
- Avoid I/We/Our throughout the manuscript. Instead, use “The present/current study”.
We change all expressions we/ our as you suggest.
- Revise the objective in Line 15. “The present study aimed to evaluate the effect of Isoginkgetin on U87MG glioblastoma cell migration.
We revised in the test the objective in Line 15
- ISO or Iso, make it uniform throughout the manuscript.
We have replaced and approved Iso in all the mentions, we thank you for letting us know.
- In the abstract, describe the conclusion in more detail and offer suggestions for the reader. so that readers will understand the overall purpose of this research after reading the abstract.
We change the abstract and eliminated suggestions to stimulate the complete reading.
- Only two of the introduction's references were recently published (10 and 11). Particularly when discussing "Glioblastoma," the introduction section needs to be updated with the most recent references that have been published in the last five years (12, 13, and 14).
We have updated the reference and thank you for the suggestion
- Line 65-69, revise it. The readers will have trouble understanding the objective.
We revised the test line 65-69 in Treatment with Isohas a double effect, it blocks the growth of glioblastoma cells and inhibits their migration.
- In figure 2, there was no ** p-value and *** p-value, please check the figure legends and remove them.
We removed in fig 2 p-value.
- Similarly, in figure 4, there was no *** p-value, please check the figure legends and remove it.
We removed p-value also in fig. 4.
- The writing in the results section is excellent. Another issue I observed was the lack of explanation in the discussion. Some results were mentioned again in the discussion. Minimize it. I would advise the authors to refocus their discussion on each section rather than adding more background information about the literature so that it is clear how the research's findings fit into the overall framework of what is happening with glioblastoma at the moment. The methods section was well-detailed. I appreciate it. The discussion section has been completely revised, eliminating the repetition of results and inserting the current state of the art for glioblastoma treatment.
- I advise authors to break up the conclusion section and provide a critical defense of their findings and observations. As a result, everyone will recognize the significance of this study. The conclusion must also address potential points of view. The author should stress the significance of this study piece.
We added a separate section for the conclusion: Considering the extreme intra- and intertumoral heterogeneity of glioblastoma and the innate or acquired resistance to conventional therapy, the present study aimed to evaluate the effect of ISO, a natural substance, a single agent on the growth of U87MG, on the clonogenic potential and/or migration. Furthermore, an attempt was made to identify the main mechanisms underlying growth inhibition, highlighting that Iso affects cell blocking and migration by initiating apoptosis and autophagy. The most interesting aspect, of course; could be investigated in a future study, whether ISO has the ability to act synergistically with temozolomide by bypassing drug resistance mechanisms, causing conventional therapy to fail and reactivating homeostatic control in glioblastoma cells.
Reviewer 3 Report
The manuscript describes the natural compound- Isoginkgeting as an agent which controls U87MG glioblastoma cell growth and migration
I have the following suggestions and concerns about this manuscript.
1. Generally, in the Supplementary data and the main manuscript the authors put the same information. I do not understand why? This should be changed.
2. Please, explain :
How was calculated IC50?, How was calculated the standard deviation (SD)? The obtained values of SD for IC50 are too high. How many experiments were done?
3. The figures need correction, there are too many drawings on each one and they are not clear. Some of the data should be placed in the supplementary data and the rest separated into individual figures.
For examples:
Ad Fig 1 – the structure of Iso I suggest putting to the Introduction, Fig 1B, 1C, 1E and IF I suggest removing and put to the Supplementary data and then leaving the rest but as the separate figures
Ad Fig 2 – Fig A I suggest putting to the Supplementary data and leaving only Fig 2B with appropriate caption,
Ad Fig 3 and 4 similarly, I suggest removing pictures and putting them to the Supplementary data and leaving graphs as separate figures with appropriate captions
4. Figures should clearly and briefly described, what they represent, while the interpretation of the results and their discussion should be in the text in the Results section and not under the figures.
5. The signature Fig 3 appears twice (lines137 and 180)
6. Lines 123 -124 “ and ERK1 / 2 protein, after treatment with iso at different times () In Fig.3 D/ E The figure shows the results of the western blots in” are not clear, please check and correct
7. The discussion seems to be too long, it should contain interpretations of the obtained results and conclusions drawn from the research. The description of the literature reports (lines191-197, 198-210, or 252-267) is too long and should rather be included in the Introduction section or be shorter.
Author Response
The manuscript describes the natural compound- Isoginkgeting as an agent which controls U87MG glioblastoma cell growth and migration
I and the other authors of the paper thank you for appreciating our study and for giving us valuable suggestions to further improve the presentation of our data:
- Generally, in the Supplementary data and the main manuscript, the authors put the same information. I do not understand why? This should be changed.
We did not add supplementary data.
- Please, explain: How was calculated IC50?, How was calculated the standard deviation (SD)? The obtained values of SD for IC50 are too high. How many experiments were done?
IC50 was calculated by doing an MTT assay with increasing concentrations of Iso. The MTT assay was done twice. The data processing was done with a particular GraphPad function that calculates IC50 and SD at 24-48 and 72h and plots the graph.
- The figures need correction, there are too many drawings on each one and they are not clear. Some of the data should be placed in the supplementary data and the rest separated into individual figures.
For examples:
Ad Fig 1 – the structure of Iso I suggest putting to the Introduction, Fig 1B, 1C, 1E and IF I suggest removing and put to the Supplementary data and then leaving the rest but as the separate figures
Ad Fig 2 – Fig A I suggest putting to the Supplementary data and leaving only Fig 2B with appropriate caption,
Ad Fig 3 and 4 similarly, I suggest removing pictures and putting them to the Supplementary data and leaving graphs as separate figures with appropriate captions
In our opinion the figures are well representative and exhaustive of the results presented in the paper, we would prefer to leave them as presented in the initial version.
- Figures should clearly and briefly describe, what they represent, while the interpretation of the results and their discussion should be in the text in the Results section and not under the figures.
We simplified the fig.4 legend as suggested.
- The signature Fig 3 appears twice (lines137 and 180).
We have corrected the text, we apologize, in lane 180 there is an error the correct wording is Fig. 4.
- Lines 123 -124 “ and ERK1 / 2 protein, after treatment with iso at different times () In Fig.3 D/ E The figure shows the results of the western blots in” are not clear, please check and correct.
Thank for your suggestion we checked and corrected the figure.
- The discussion seems to be too long, it should contain interpretations of the obtained results and conclusions drawn from the research. The description of the literature reports (lines191-197, 198-210, or 252-267) is too long and should rather be included in the Introduction section or be shorter.
The discussion was totally revised, thanks for all your suggestion
Round 2
Reviewer 1 Report
The author has made some improvements to the article, but the changes to the figures are not satisfactory. The scale in the figures is still not clear enough. I would suggest that the author continue to improve them.
Reviewer 2 Report
I really appreciate the authors' effort because they made significant changes to the manuscript in response to my comments. As a result, I recommend considering it for publishing in the Molecules Journal in present form.
Reviewer 3 Report
I accept in present form.